# Thai Oakleaf Lettuce Phenocopies a Phytochrome B Mutant

**DOI:** 10.3390/biology13060390

**Published:** 2024-05-28

**Authors:** Cade Cooper, Kevin M. Folta

**Affiliations:** Horticultural Sciences Department, University of Florida, Gainesville, FL 32611, USA

**Keywords:** lettuce, *Lactuca sativa*, phytochrome, controlled environment

## Abstract

**Simple Summary:**

Lettuce is an important crop in indoor production (“vertical farming”) systems. In the current study, seedling growth habits were analyzed to make predictions about mature plant growth and development under narrow bandwidth light. We show that the lettuce variety Thai Oakleaf demonstrates defects in normal red-light-mediated development, which is reminiscent of mutations in a central red light photosensory pathway.

**Abstract:**

Photomorphogenic development in seedlings may be diagnostic of future plant performance. In this report, we characterize the Thai Oakleaf lettuce genotype, as it exhibited abnormalities in photomorphogenic development that were the most conspicuous under red light, including defects in hypocotyl growth inhibition, decreased cotyledon expansion, and constitutive shade avoidance tendencies. These observations are consistent with defects in red light sensing through the phytochrome B (phyB) photoreceptor system. This genotype is sold commercially as a heat-tolerant variety, which aligns with the evidence that phyB acts as a thermosensor.

## 1. Introduction

Indoor farming, or controlled environment agriculture (CEA), offers an opportunity to produce high-value horticultural crops in proximity to major population centers where they are consumed. The potential benefits include a shorter postharvest time until it reaches the consumer, along with a lower carbon footprint and the ability to repurpose urban infrastructure towards food production. One of the central limitations of CEA is the high cost of energy, which can be over 25% of the operational costs [1].

One area of potential improvement is the plant itself. There are many examples where crop plants may possess loss- or gain-of-function alleles in light sensing hardware that affect their desirable traits [2]. It is possible that existing varieties may possess alleles in photosensory systems that render them more amenable to growth in an artificial low-light environment. Efforts to identify these genotypes could lead to the identification of more productive cropping varieties, plants that produce consumer-facing traits in CEA, or germplasm well suited as parents for breeding cultivars optimized for CEA production [3].

Plants use light for photosynthesis, but light wavebands are also relevant in establishing patterns of gene expression and physiology that best match the plant to prevailing light conditions. A series of light sensory systems detect energy spanning the visible spectrum and its adjacent regions. A family of phytochrome receptors mediate responses to the red and far-red portions of the spectrum [4,5]. Blue light/UV-A responses are controlled by the cryptochrome [6] and phototropin [7] family of receptors. UV-B is sensed through a receptor known as UVR-8 [8], and unusual low-light responses occur in all photosensory receptor mutants, suggesting that they are transduced via a yet-to-be-described sensory system [9,10]. A controlled light environment allows for the identification of genotypes that align well with a specific ambient spectrum based on their potential to signal through these discrete environmental input channels.

The developing seedling is an outstanding system to identify variations in light sensitivity between genotypes. The dark-grown seedling rapidly displays a number of conspicuous and readily quantifiable traits upon transfer to light. An early hypocotyl growth rate, cotyledon expansion, and the accumulation of pigments like chlorophyll have been traditionally used to study the integration of light signals through defined photosensory systems. An evaluation of these traits in developing seedlings may provide information about the sensitivity to light, which could potentially be extrapolated to mature plants. The experiments are rapid and inexpensive and require minimal space to perform, yet they are highly informative about the relationship between a given lettuce variety and the light environment.

In a broad survey of lettuce varieties, the Thai Oakleaf variety exhibited growth defects when grown under red light, consistent with decreased phytochrome signaling. Defects in light sensing were not observed for other wavelengths. This variety is characterized in detail in this report, demonstrating defects in red light sensing during seedling and early maturity.

## 2. Materials and Methods

### 2.1. Plant Materials

Experimental seeds were obtained in at least two individual batches. Two separate sets of ‘Thai Oakleaf’ lettuce seeds were obtained from Southern Exposure Seed Exchange (Mineral, VA, USA) and Seedwise (Folsom, CA, USA) with comparable results from each. Other comparator varieties were obtained from Johnny’s Select Seeds (Winslow, ME, USA). Reference varieties were chosen based upon their growth characteristics in survey experiments (Appendix A, Appendix A) as they best approximated the mean of all seedlings under the wavebands and fluence rates tested.

### 2.2. Light Sources

Light was provided from custom-built LED arrays where fluence rate and light combinations were modulated by pulse wave modulation via a Raspberry Pi (Cambridge, UK) based controller. Continuous blue LED light was provided with peak wavelength at 470 nm, red light peak at 660 nm, and far-red light peak at 730 nm. White light was generated by cool white fluorescent bulbs.

### 2.3. Early Photomorphogenesis Experiments

For hypocotyl growth inhibition experiments, seeds were plated in a line on square Petri plates containing 1% agar with 1 mM CaCl_2_ and 1 mM KCl as described [11]. Seeds were stratified for 2 days at 4 °C, warmed to room temperature, and then illuminated with white fluorescent light (~20 µmol/m^2^s) for 1 h. The plates were then moved to experimental conditions, standing vertically beneath the light source. Seedlings were exposed to red light at 10, 30, and 50 µmol/m^2^s or blue light at 0.5, 5, and 50 µmol/m^2^s. Dark controls were comparable plates wrapped in aluminum foil in parallel chambers. Seedlings were grown for 96 h and then imaged. Quantitative data were obtained by measuring hypocotyls with ImageJ (https://imagej.net/ij/) using the gridlines on the plates as a size standard. Experiments were performed at least three times. Means were analyzed using one-way ANOVA with Tukey’s HSD at ά = 0.05.

Cotyledon expansion was quantified using a similar approach, and only Petri dishes were placed perpendicular to ambient light. Seedlings were imaged after 96 h, and areas were analyzed using ImageJ as described above. Leaf area was measured on the second true leaf to emerge on young plants after two weeks on the same media under various wavelengths and fluence rates. The leaf area was calculated using Image J calibrated by appropriate size standards. Means were analyzed using one-way ANOVA with Tukey’s HSD at ά = 0.05.

### 2.4. Shade Avoidance

Seedlings germinated on 1% agar plates were transferred to soil-less mix in 23 cm^−2^ baking pans and then moved to light conditions. All light conditions included blue light at 50 µmol/m^2^s, and red/far-red ratios were varied. Red light alone was supplied at 50 µmol/m^2^s. Red + far red was supplied at 50 µmol/m^2^s each, and the 1:10 red/far-red treatment concentrations were 5 µmol/m^2^s red and 50 µmol/m^2^s far red. Plants were grown for 21 days and then imaged. Experiments were repeated three times with similar results.

## 3. Results

### 3.1. Fluence Rate/Response Experiments

A survey of many lettuce genotypes showed that the ‘Thai Oakleaf’ seedlings exhibited less inhibition of the hypocotyl elongation rate under red light, so the response was investigated further. ‘Thai Oakleaf’ was compared to a series of other genotypes that were selected for their average response to red, blue, or white light (a subset presented in Appendix A). To test the basis of the phenotype, the seedlings were grown under three fluence rates of light, as described in the Section 2. Images of individual seedlings are presented in Figure 1 (panel A), and the quantitative data from many individual replicates are shown in panels (B–D). The dark growth rates were comparable across genotypes.

The inhibition of hypocotyl elongation was impaired in Thai Oakleaf under 10, 30, and 50 µmol/m^2^s of red light relative to the controls (Figure 1B). The results show that all seedlings exhibit strong inhibition of the hypocotyl elongation rate under all fluence rates of blue light, with no significant differences observed between genotypes (Figure 1C). White light generated comparable inhibition across genotypes, except at 5 µmol/m^2^s, where Thai Oakleaf (TOL) and Summer Crisp (SC) showed less inhibition.

Cotyledon expansion is another hallmark of photomorphogenic development. Thai Oakleaf lettuce only exhibited significant inhibition of cotyledon expansion under red light compared to the other light qualities tested at 50 µmol/m^2^s for 96 h (also visible in Figure 1A). To quantify the response, many seedlings were grown under different light qualities for 96 h and then imaged from above. Their cotyledon areas were calculated. Images of single seedlings are shown in Figure 2A, with the results of many seedlings presented in Figure 2B.

### 3.2. True Leaf Growth

Leaf expansion is light mediated, and the leaf size, shape, and color are important commercial aspects of lettuce production. It was therefore of interest to test whether the red light sensing defects observed in the seedlings were apparent beyond the cotyledon stage. Thai Oakleaf plants were grown for two weeks under increasing fluence rates of blue, white, or red light. The results show that leaf expansion was affected by the fluence rate, with blue and white light leading to significant changes in the leaf area. White light was not as effective as blue light at the highest fluence rate. There was no significant difference between various fluence rates of red light (Figure 3).

### 3.3. Responses to Variation in R/FR Ratio

In Arabidopsis, mature *phyB* mutant plants show some tendencies toward a shade avoidance phenotype under low light conditions, as phyB is the central receptor dictating the response to R:FR conditions. The effects of changing ratios of red and far-red were examined in Thai Oakleaf to test whether the apparent *phyB*-like defects were observed in more mature plants. Plants were grown under various combinations of blue light with various ratios of red and far-red added. The results are presented in Figure 4. Figure 4A illustrates the elongated leaf growth observed in Thai Oakleaf compared to two other similar varieties, Oscarde Oakleaf and Red Sails. All treatments contained blue background light to limit the effects of shade responses to low blue conditions [12]. The results show that Thai Oakleaf lettuce presents a longer leaf (petiole + leaf blade) compared to the other varieties. Far-red light augments the leaf length, especially at a low red/far-red ratio. Examples of leaf angles in Thai Oakleaf are illustrated in Figure 4B where the red/far-red ratio is 1 and where the red/far-red ratio is 0.1, where Thai Oakleaf presents a shade avoidance phenotype, noted by a lower angle of leaf inclination, even at an equal red/far-red ratio.

## 4. Discussion

Plant photosensors control many aspects of plant growth, development, morphology, and metabolism in response to signals from the ambient environment. In a horticultural context, many of these responses ultimately affect the crop performance or product qualities that bring value to the industry and/or consumer [2]. Because light plays such a central role in shaping the horticultural plant stature and quality, it is not surprising that specific land races and cultivars were selected because of loss- or gain-of-function in light sensing pathways that affected a desirable trait.

This phenomenon has been described in crop plants as well as model systems. For instance, the Cape Verde Islands accession of *Arabidopsis thaliana* is a *cryptochrome*2 gain-of-function allele that affects seedling traits and temperature sensitivity [13,14,15]. The overexpression of phyA induced dwarfing and increased production in rice [16], while overexpression of the Cry2 receptor in tomato led to higher fruit pigmentation and lycopene content [17]. Light sensors are also central in controlling the flowering time [18], and they play a central role in partitioning resources in light-limited environments [19], making them attractive targets for breeding in controlled environments.

From this foundation, we test the hypothesis that the light sensory variations may provide advantages (or disadvantages) that are especially conspicuous when crops are grown under artificial lighting. Rapid, simple, and inexpensive assays in developing seedlings may reveal these sensory variants and allow for the prioritization of varieties to test, as well as a description of the wavebands that affect their growth and/or development. These varieties and their resident mutations may then be integrated into (or excluded from) breeding programs aimed at developing varieties specifically for controlled environments.

A yet unpublished survey of many lettuce varieties under different light conditions revealed that the Thai Oakleaf variety exhibited developmental impairment under red light. The phenotypes are consistent with mutations in the phytochrome B (phyB) signaling pathway [20]. Under red light, the plants showed less hypocotyl elongation inhibition under all fluence rates tested, leading to taller seedlings under these conditions. Significant differences were not observed under any fluence rate of blue light, and smaller differences were observed under white light with a low fluence rate (0.5 µmol/m^2^s), which is consistent with *phyB* mutations in Arabidopsis [21]. This finding is not surprising because “white” LED conditions generated by fluorescent bulbs or LEDs are a combination of red, blue, and green emissions that are perceived by the human eye as white. The plant is experiencing the co-activation of multiple sensory systems under this illumination, and the inability to sense red correctly manifests as the difference in elongation suppression observed. At higher fluence rates, the blue sensory pathways can apparently compensate for the loss of red light activation.

The same trends are observed in cotyledon expansion, as white and blue light treatments lead to increases in the surface area that are easily discernable in Thai Oakleaf as well as reference genotypes. However, the Thai Oakleaf variety is relatively insensitive to red light, and cotyledons remain almost perfectly closed under red light treatments with a high fluence rate (Figure 2). The expansion of true lettuce leaves is driven by both blue and red light [22,23]. The results in Figure 3 show that increasing the fluence rates of blue light results in larger leaves. Red light has little effect on leaf expansion, and white light is impaired, most likely because the white light source is a combination of red and blue wavebands, so the observed expansion likely represents a response to blue alone. These findings are also consistent with a phyB signaling mutation. Significant differences are also observed between the Thai Oakleaf and reference genotypes for white light; again, this is likely due to the fact that “white” light contains red wavebands.

While seedling phenotypes are informative, it is important to test how they translate to plants through development. In this case, it was of interest to determine whether the *phyB*-like defects observed in seedlings translate to more mature plants. The phyB sensor is central to interpreting the ratio of red to far-red light, measuring the proximity to neighboring plants as well as tailoring the morphology and physiology for growth under a canopy of leaves [20,24]. In an enhanced far-red environment, plants take on a more erect stature with greater leaf inclination, longer leaves, and a change in the ratio of petiole to leaf blade, known collectively as shade avoidance syndrome (SAS). Because far-red light moves phyB to its inactive form, mutations in phyB show some degree of shade avoidance phenotypes [24]. The results in Figure 4 show that Thai Oakleaf lettuce shows features that are consistent with defects in red light sensing through phyB. Panels 4A and B show that Thai Oakleaf has longer leaves that are more erect, reminiscent of a shade avoidance response. The response is present in the absence of far-red light, but it increases with the gradual addition of far-red light, and it is prominent under low ratios of red to far-red. While all varieties exhibited some response to the enhanced far-red environment, Thai Oakleaf exhibited a more prominent response, which is consistent with phyB signaling defects.

While most likely to be a defect in phyB, other Type II phytochromes have also been shown to produce similar phenotypes in seedling development and SAS [25]. The defect may also reside downstream of phyB, such as in its nuclear import or direct connections to gene expression. Mutations in the *HEME OXYGENASE 1* gene result in defects across all phytochrome receptors [26], leaving the plants unable to properly perceive red or far-red light due to the inability to synthesize sufficient phytochrome chromophore [27]. However, it is not likely that the basis of the phenotypes is here because the seedling response to far-red light through phyA was normal (Appendix A).

The reference lettuce genome (cv. Salinas) contains a single copy of *PHYB*, residing on chromosome 1. A single lesion could potentially disrupt phyB function. However, it has been reported that lettuce phyB is expressed as several splice variants in response to heat stress [28]. Further analyses that examine the gene and its splice variants may reveal the molecular basis of the observed phenotypes in red light.

There is little information available about ‘Thai Oakleaf’ lettuce and its selection. It is reported to be a cultivar from a breeding program in Thailand and was selected for heat tolerance [29]. Retailers note that it has an upright growth habit and is slow to bolt. These qualities are also consistent with a lesion in phyB, which has been described as a thermal sensor as well as a red light photosensor [30,31]. The PHYB protein has been shown to directly interact with promoters of temperature-regulated genes in darkness, turning them off [30]. It is therefore consistent that Thai Oakleaf, selected for its high-temperature tolerance, would leave genes associated with acclimation to warm temperatures de-repressed, potentially underlying its tolerance to high temperatures.

The limitations of the present work are that the experiments were confined to the seedling and young leaf stages, and long-term effects on the commercial product remain to be tested. The later leaf expansion and shade avoidance effects, while consistent with defects in phyB sensing, may be due to factors independent of light, as many leafy greens (e.g., romaine lettuce) are selected for an upright stature. Future experiments will examine the relationship between early phenotypes and long-term performance in CEA under varying fluence rates and ratios of red/far-red light.

## 5. Conclusions

The ‘Thai Oakleaf’ lettuce genotype exhibits aberrant growth habits that are consistent with a role in phyB-mediated light sensing. On a larger level, these findings underscore the utility of identifying potential heat-tolerant or CEA-favorable genotypes through an analysis of seedling traits using relatively inexpensive and rapid growth and development assays to speed up selection and long-term breeding efforts. 

## Figures and Tables

**Figure 1 biology-13-00390-f001:**
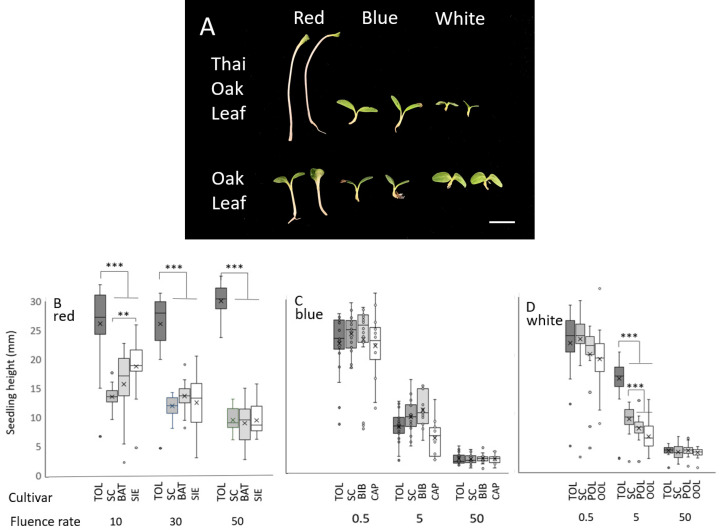
Thai Oakleaf lettuce lacks normal response to red light in developing seedlings. Hypocotyl lengths of seedlings grown under red, blue, or white light are presented. Panel (**A**) represents Thai Oakleaf and Pannisse Oakleaf seedlings grown under 50 µmol/m^2^s of red, blue, or white light for 96 h. Line = 1 cm. Quantitative data from 15–25 seedlings grown under 10, 30, and 50 µmol/m^2^s red light are presented in panel (**B**). Panels (**C**,**D**) present data from 15–25 seedlings grown under blue or white light, respectively, at 0.5, 5, and 50 µmol/m^2^s. TOL, Thai Oakleaf; SC, Summer Crisp Butterhead; BAT, Batavian Crisphead; SIE, Sierra Crisphead; BIB, Speckled Bibb; CAP, Capitan; POL, Pannisse Oakleaf; OOL, Oscarde Oakleaf. Means were analyzed using ANOVA with Tukey’s HSD; *** = *p* < 0.001, ** = *p*< 0.01.

**Figure 2 biology-13-00390-f002:**
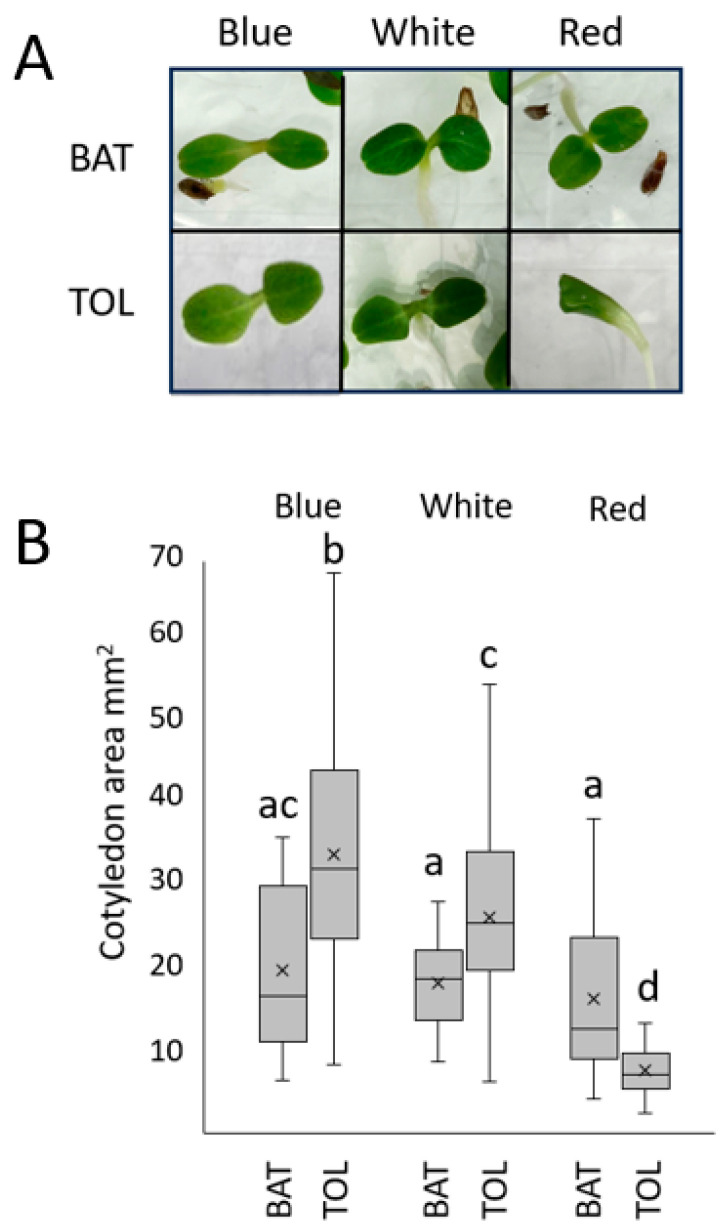
Thai Oakleaf lettuce exhibits defects in red light-mediated cotyledon expansion. Seedlings were grown under 50 µmol/m^2^s blue, red, or white light for 96 h. Panel (**A**) shows examples of individual cotyledon pairs showing expansion (not to scale), and (**B**) shows quantitative results from at least 22 seedlings. BAT—Anuenue Batavian Lettuce; TOL—Thai Oakleaf. Means were analyzed using ANOVA with Tukey’s HSD pairwise comparisons; letters denote non-significant differences between treatments.

**Figure 3 biology-13-00390-f003:**
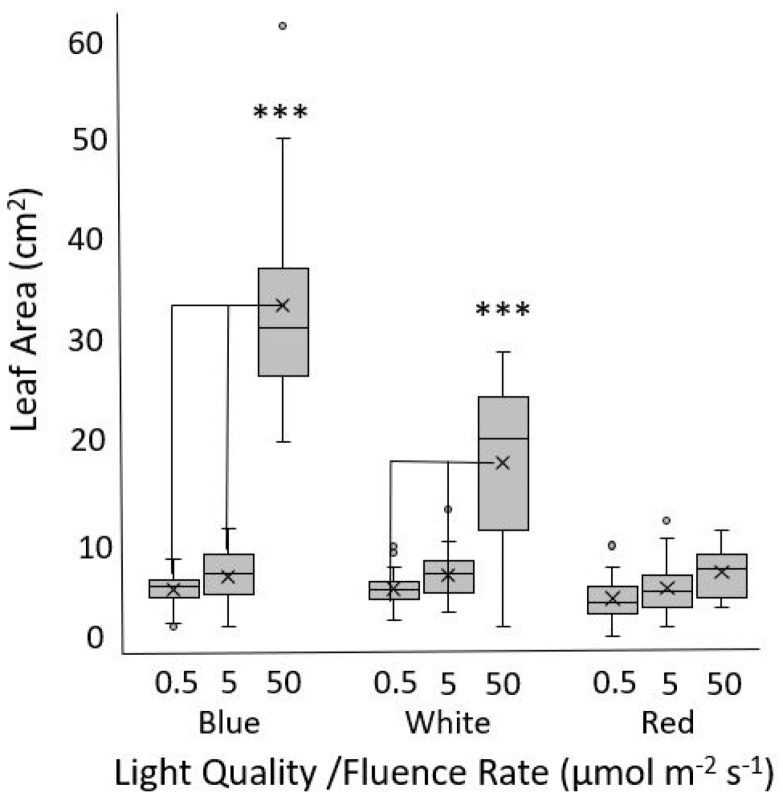
Leaf expansion in response to various fluence rates and wavelengths of light. Thai Oakleaf plants were grown for 14 d under three fluence rates of blue, white, and red light, and the area of the second true leaf was measured. At least 20 plants were measured per experimental replicate, and similar results were obtained over three separate experiments. Significant differences are noted by triple asterisk (***) as determined by ANOVA with Tukey’s HSD at *p* < 0.001.

**Figure 4 biology-13-00390-f004:**
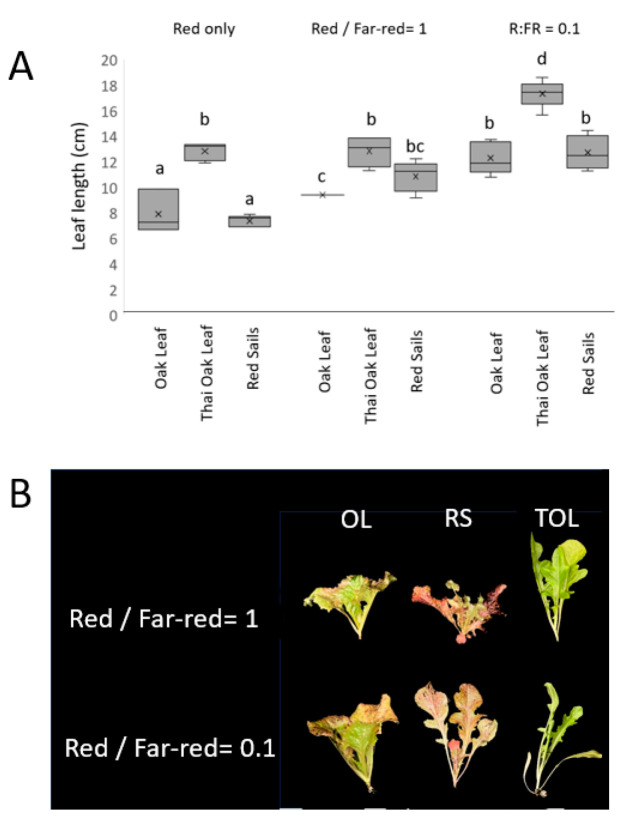
Changes in plant morphology induced by variations in red/far-red ratio. Plants were grown for 21 d under constant blue light (50 µmol/m^2^s) supplemented with varying ratios of red (50 or 5 µmol/m^2^s) to far red light (50 µmol/m^2^s). Panel (**A**) shows length of second true leaf (petiole plus midrib) in Thai Oakleaf, Oscarde Oakleaf, and Red Sails varieties in response to treatment with red alone, red, and far-red at equal amounts and red/far-red ratio of 0.1. Letters represent significant pairwise differences using ANOVA and Tukey’s HSD, *p* < 0.05. Panel (**B**) shows representative plant under two ratios of red and far-red light.

## Data Availability

The data obtained in these trials are freely available upon request.

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
