# Peer review of "Thai Oakleaf Lettuce Phenocopies a Phytochrome B Mutant"

_biology, 2024, doi:10.3390/biology13060390_

Round 1

Reviewer 1 Report

Comments and Suggestions for Authors

Comments to authors

The authors studied the seedling growth habits of Thai Oak Leaf lettuce and make predictions about mature plant growth and development under narrow bandwidth light. The work seems interesting of but there are certain issues which the authors have to address before accepting the manuscript (MS) for publication, 

1.     Leaf area was measured on the second true leaf to emerge on young plants after two weeks on the same media, under various wavelengths and fluence rates.  Kindly explain this as it looks very confusing.

2.     Thai Oak Leaf lettuce exhibits defects in red light mediated cotyledon expansion. Seedlings were grown under 50 µmol/m2s blue, red, or white light for 96 h. Why cotyledons expansion studies performed under only 50 µmol/m2s without any variations?

3.     What are the possible molecular or physiological mechanisms causing the observed plant responses to changes in R/FR ratio? Any specific signalling pathways or gene regulatory networks involved?

4.     The results show that leaf expansion was affected by fluence rate, with blue and white light leading to significant changes in leaf area. Why is blue light more effective?  Also the authors did not incorporate Figure 3 in any parts of the MS. They show only in the figure section. Kindly put the figure in the body of MS.

5.     The discussion part lacks mention of related works conducted previously. Incorporate the latest works in this section. The authors should mention whether the present findings align with or different from previous works on plant responses to light quality.

Author Response

Plase see attachment

Reviewer 2 Report

Comments and Suggestions for Authors

In Cooper and Folta, the authors examine the responses to red and far red light in the Thai Oak Leaf lettuce genotype. The authors identify traits of Thai Oak Leaf’s responses that are consistent with defects in phytochrome B.

At several points, the authors make reference to unpublished datasets that are not shown or included in the manuscript. The omission of these datasets make it impossible to judge some of the claims made by the authors. For example, in Plant Materials L69 the authors make reference to growth assays of several lettuce varieties which identified Thai Oak Leaf as displaying an interesting phenotype. In the Discussion L180-181, they further reference a “yet unpublished survey of many lettuce varieties”, presumably the same study. The authors should either publish this foundational study before making claims base don unpublished datasets, or at least provide the manuscript of the unpublished survey as an additional document for review purposes only. Without at least the latter, it is impossible to judge how Thai Oak Leaf behaves relative to the other varieties being assayed and if the developmental impairments under red light point to Thai Oak Leaf as a clear candidate for further study as claimed by the authors.

Similarly, the authors propose an alternative hypothesis for the cause of the phenotypes observed in Thai Oak Leaf, only to write off this alternative hypothesis based on far red light responses that are not shown in the manuscript. These data are critical for the authors’ argument that the TOL variety displays a phyB-only phenotype, and yet the data are not shown. Why? If these data really support this argument and the authors want to make this argument (that this phenotype is phyB-specific), these data need to be added to the manuscript.

In addition, in the Discussion L228 the authors make reference to personal communications. References to personal communication are antiquated and generally no longer accepted as valid in most peer-reviewed literature, for similar reasons as the above “data not shown” cases. If there is information to demonstrate this, the authors should cite it, or at least point out that there is not clear data/no way currently to verify this claim.

In addition, some of the method descriptions are unclear/insufficient. In L93, which wavelengths and fluence rates were used? The results send the reader to the methods to discover these numbers, but they should be provided here clearly. As it stands, it is unclear where this information is meant to be found by the reader. Similarly in the “shade avoidance” section L99, where are the ratios specified? There is not even a mention to the part of the results where this information is presumably supposed to be found. Anyways, these ratios should be listed here in the methods. In the Results L108, it is not clear enough what is meant by there three fluence rates of light. The specific Materials and Methods section should be referred to. Is this referring to the “early photomorphogenesis experiments” part of the methods? If so, it is not clear what the three fluence rates are from this section.

Some of the results descriptions are either inadequate or missing information (e.g. significance markers in Figs. 2 and 4). In L115-117, the results could use some clarification. Do the authors mean e.g. that SC had a significantly different growth rate relative to the other Oak Leaf varieties? Here and elsewhere, “a significant difference” does not tell us what is significantly different about TOL or SC relative to the other varieties (presumably, hypocotyl elongation rate or length). This should be made clearer. In addition, it would be useful to provide a clear vector of difference (i.e. “significantly longer/shorter hypocotyl lengths relative to other varieties”). In L122, the authors say “significant inhibition” but Figure 2 doesn’t show any significant differences. If this is an oversight on the figure, this should be addressed, but a visual inspection doesn’t inspire confidence that these means would actually be significantly different according to the test used in the other figures (ANOVA followed by Tukey HSD). If this is the case, the authors should change this language to discuss the trends, and not say “significant” when they are not statistically significantly different.

In Figure 1, why are some lines used for red light, others for blue, and yet different ones for white light? Only Thai Oak Leaf and Summer Crisp seem to be consistent. Presumably this has to do with the data not shown mentioned in the results section, but the authors should discuss these data (shown in a supplemental figure) and provide a rationale for why only these lines were used for each of those conditions, if they are not going to show all lines under all conditions.

As in Figure 2, the authors imply the results in Figure 4 are significantly different (e.g. L151, “presents a longer leaf”), but there are no significance markers in Figure 4. Are these results significant or not? If they are not significant, the authors should not say that TOL leaves are longer in Figure 4, and instead discuss trends. However, if half the results (figs. 3 and 4) do not demonstrate significant results and only trends despite adequate replication, this might call into question  the conclusions of the study.

Finally, in the Discussion e.g. L196-197 and thereabouts, the authors discuss that the TOL phenotypes are consitent with a phyB mutation. In several species, such as maize (see Sheehan et al. 2007 Plant J) and Nicotiana attenuata (Fragoso et al. 2017 JIPB), there are multiple copies of phyB that correspond to different aspects of the canonical phyB phenotype. Is it known if phyB has only one copy in lettuce? If this is not known, this possibility should be discusses and the phenotypes considered in light of this.

Comments:
L15: photochrome -> phytochrome
L18: “and also that” maybe “with the fact that” or something similar?
L26: “carbon footprint, and ability” -> “carbon footprint and the ability” (remove comma, add “the”)
L94: remove space between Image and J
L125: show -> shown
L127-134: Some references to Figure 3 should be made in this paragraph where appropriate
Figure 3: remove “light quality” from the x-axis (only fluence rate, which is what is being shown/measured)

Comments on the Quality of English Language

No particular comments/minor comments included in comments to authors (English quality is generally good)
